# SARS-CoV-2 Infection and Pancreatic β Cell Failure

**DOI:** 10.3390/biology11010022

**Published:** 2021-12-24

**Authors:** Keiichiro Mine, Seiho Nagafuchi, Hitoe Mori, Hirokazu Takahashi, Keizo Anzai

**Affiliations:** 1Division of Metabolism and Endocrinology, Department of Internal Medicine, Faculty of Medicine, Saga University, 5-1-1, Nabeshima, Saga 849-8501, Japan; su2733@cc.saga-u.ac.jp (S.N.); sunrisebyebyemoon@yahoo.co.jp (H.M.); takahas2@cc.saga-u.ac.jp (H.T.); akeizo@cc.saga-u.ac.jp (K.A.); 2Division of Mucosal Immunology, Research Center for Systems Immunology, Medical Institute of Bioregulation, Kyushu University, 3-1-1, Maidashi, Fukuoka 812-8582, Japan; 3Liver Center, Saga University Hospital, Saga University, 5-1-1, Nabeshima, Saga 849-8501, Japan

**Keywords:** virus, diabetes, COVID-19, SARS-CoV-2, pancreas, β cell

## Abstract

**Simple Summary:**

Accumulating evidence suggests that the severe acute respiratory syndrome coronavirus 2 (SARS-CoV-2) may have the potential to induce pancreatic β-cell damage, leading to diabetes onset in patients with coronavirus disease 2019 (COVID-19). However, controversial results have been reported among study groups. Here, we provide a comprehensive review of published findings that describe the potential relationship between SARS-CoV-2 infection (COVID-19) and pancreatic β-cell failure, and how this may contribute to the development of diabetes.

**Abstract:**

SARS-CoV-2 infection primarily causes pulmonary symptoms; however, accumulating reports indicate that some patients with COVID-19 have multiple organ dysfunction or failure. Although diabetes is considered a risk factor for severe COVID-19, SARS-CoV-2 infection may also be a causal factor for diabetes mellitus in patients with COVID-19. According to the research reviewed in this paper, the pancreas and pancreatic β cells appear to be targets of SARS-CoV-2 and are damaged by direct or indirect effects of the infection. However, controversial results have been reported between study groups, mainly due to the limited number of cases with diabetes precipitated by COVID-19. In this review, we comprehensively discuss the published findings on the potential association between SARS-CoV-2 infection or COVID-19 and pancreatic β-cell damage leading to diabetes onset. These findings will further contribute to our understanding of the pathogenesis of diabetes mellitus.

## 1. Introduction

SARS-CoV-2 is a coronavirus that was first identified in the city of Wuhan, Hubei Province, China, in late 2019, and is the causal agent of the global COVID-19 pandemic [1]. SARS-CoV-2 belongs to the Coronaviridae family, and is an enveloped virus with a single-strand, positive-sense ribonucleic acid (RNA) genome [2,3]. Similar to the severe acute respiratory syndrome (SARS) and Middle East respiratory syndrome (MERS) outbreaks, it has been suggested that COVID-19 was initiated by zoonotic spillover and is now transmitted between humans [4]. Although SARS-CoV-2 infection primarily causes pulmonary manifestations such as cough, dyspnea, pneumonia, and acute respiratory distress syndrome, accumulating reports show that multiple organ dysfunction or failure is evident in some patients with COVID-19. Persistent symptoms and/or long-term complications (sequelae) may also be evident in the post-acute COVID-19 phase, 4 weeks after the onset of acute symptoms of COVID-19, even when the virus is undetected by a polymerase chain reaction (PCR) test [5,6]. Therefore, COVID-19 is a multi-organ disease with a broad spectrum of manifestations [5].

Potential risk factors for higher severity and mortality among COVID-19 patients have been reported: older age, higher Sequential Organ Failure Assessment (SOFS) score, elevated D-dimer, hypertension, cardiovascular disease, kidney disease, obesity, and diabetes [7,8,9,10]. Diabetes itself does not appear to increase the risk of COVID-19 infection; however, patients with diabetes mellitus type 1 and 2 have a poorer prognosis [9,11,12]. In the Scottish population, as of 31 July 2020, the odds ratio (OR) for the development of severe COVID-19 in patients with diabetes was 1.395 (95% confidence interval (CI) 1.304–1.494; *p* < 0.0001) compared with those without diabetes [9]. The OR was 2.396 for type 1 diabetes (1.815–3.163; *p* < 0.0001) and 1.369 for type 2 diabetes (1.276–1.468; *p* < 0.0001) [9]. Diabetes is associated with worse outcomes in patients with COVID-19, likely as a result of diabetic complications, such as hypertension and cardiovascular disease, which are independent risk factors for COVID-19 [11]. In addition, it is thought that the dysregulated immune response of airway epithelia due to poorly controlled blood glucose may be associated with COVID-19 severity [13]. Fortunately, improved glycemic control has been reported to be associated with better outcomes in patients with COVID-19 and diabetes [11,13].

Although diabetes is considered a risk factor for severe COVID-19, as described above, SARS-CoV-2 infection may also be a causal factor for diabetes mellitus [14,15,16]. Several viruses, including both RNA and deoxyribonucleic acid (DNA) viruses, have been described as potential causal agents for diabetes [17], such as the enterovirus [18,19,20], rotavirus [21], mumps virus [22], and cytomegalovirus [23]. In addition, during the COVID-19 pandemic, a possible association between SARS-CoV-2 infection and new-onset diabetes has been noted [11,24,25]. Viruses may induce pancreatic β-cell death or damage in several ways: induction of direct cell lysis, programed cell death, inflammation (bystander damage or bystander activation), autoimmunity against β cells, dedifferentiation or transdifferentiation, increased insulin resistance in peripheral tissues, and lipotoxicity [17,26,27,28,29,30,31]. It is well understood that interaction between host factors and viral diabetogenicity is important for the development of diabetes upon virus infection [17,26,32,33,34,35,36,37,38]. While diabetes has been described, at least in part, as a virus-induced or virus-related disease, the developmental mechanisms of viral infection-induced diabetes are complicated and still unknown [26,27]. In this review, we provide a comprehensive overview of the recent findings that describe the potential association between SARS-CoV-2 infection (COVID-19) and pancreatic β-cell damage leading to diabetes onset, to better understand the pathogenesis of diabetes mellitus.

## 2. The Properties of SARS-CoV-2

SARS-CoV-2 belongs to the Coronaviridae family and is an enveloped virus with a single-strand, positive-sense RNA genome [2,3]. SARS-CoV-2 is predicted to possess at least 12 coding regions: replicase open reading frame 1a/1b (ORF1a/1b), spike (S), 3, envelope (E), membrane (M), 7, 8, 9, 10b, nucleocapsid (N), 13, and 14 [3]. SARS-CoV-2 is genetically different to SARS-CoV (~79% similarity) and MERS-CoV (~50% similarity), which are highly pathogenic coronaviruses that cause respiratory symptoms that can be fatal. In comparison with these two coronaviruses, the SARS-CoV-2 genome encodes a longer S protein (1273 amino acids (aa); SARS-CoV, 1255 aa; MERS-CoV, 1270 aa) [3]. The S protein, which plays a role in binding to host cell receptors and viral entry, consists of a single peptide (aa 1–13) located at the N-terminus, and is composed of two subunits: S1, which harbors the receptor-binding domain (RBD), and S2, which harbors the transmembrane domain [2,39,40].

SARS-CoV-2 attaches to the host cell surface receptor, angiotensin-converting enzyme 2 (ACE2), using the RBD within the S1 subunit of the S protein [39,40,41]. Subsequent entry into the host cell requires transmembrane serine protease 2 (TMPRSS2) for S protein priming [2,39]. The other putative viral receptors and proteases that facilitate SARS-CoV-2 infection are dipeptidyl peptidase 4 (DPP4), cathepsin L (CTSL), a disintegrin and metalloprotease 17 (ADAM17), furin, transmembrane serine protease 4 (TMPRSS4), HDL-scavenger receptor B type 1 (SR-B1), and neuropilin 1 (NRP-1) [42,43,44,45,46,47,48].

SARS-CoV-2 viral load in the upper respiratory tract reaches a peak within 7 days after the onset of symptoms, which is shorter than the time taken for the peak of SARS-CoV (10–14 days) and MERS-CoV (7–10 days) [49]. However, despite the viral load peak occurring within 7 days, SARS-CoV-2 RNA shedding can be detected more than 2 weeks after the onset of symptoms. Longer term SARS-CoV-2 RNA shedding of more than 80 days has also been observed in individuals who have recovered from COVID-19 symptoms [49,50,51]. It has been reported that the persistent SARS-CoV-2 RNA shedding is not likely to be associated with COVID-19 symptom severity, circulating levels of SARS-CoV-2-specific antibodies, or SARS-CoV-2 transmission [51,52]. Older age is reportedly associated with prolonged duration of SARS-CoV-2 RNA shedding [52,53]. The role of SARS-CoV-2-specific CD8 T cell responses in prolonged RNA shedding is controversial [50,51].

## 3. A Possible Association between SARS-CoV-2, Insulin Resistance, and Diabetes

A potential association between SARS-CoV-2 infection and the development of diabetes has been revealed by clinical studies [24,25]. Marchand et al. [54] reported that a patient was diagnosed with type 1 diabetes and was positive for the glutamic acid decarboxylase-65 autoantibody one month after COVID-19 recovery. In contrast, Hollstein et al. [55] reported a case of type 1 diabetes, in which the disease had developed five to seven weeks after SARS-CoV-2 infection, without the presence of islet-related autoantibodies. Furthermore, an increased number of new-onset type 1 diabetes cases in children was observed after the initial COVID-19 outbreak in the United Kingdom [14] and Romania [56]. In addition, COVID-19 may induce diabetic ketoacidosis (DKA) and pancreatitis in some patients [57,58,59,60,61,62,63,64,65,66,67], suggesting that COVID-19 may induce pancreatic tissue damage and disrupt glycometabolic control.

Increased peripheral insulin resistance due to viral infection-induced inflammation has been reported in animal models [68]. In the context of COVID-19, increased insulin resistance has been reported even in COVID-19 patients without pre-existing diabetes [69]. These patients had an elevated triglyceride-glucose index, which is a marker of insulin resistance [69]. While the precise mechanisms of SARS-CoV-2-induced insulin resistance remain unclear, it has been suggested that complex direct and/or indirect mechanisms related to the multiple organs affected by COVID-19 may be involved. Insulin resistance is characterized by the inability of insulin to suppress hepatic glucose output, which is caused by sustained elevation of gluconeogenesis in the liver, and impaired insulin-stimulated glucose uptake in muscle and adipose tissues, which is caused by defective glucose transporter type 4 (GLUT4) cell surface expression in these tissues [70]. Despite the controversial role of inflammation in insulin resistance, inflammation caused by infiltrating immune cells in metabolic tissues has been considered as a potential factor contributing to increased insulin resistance in peripheral tissues [70]. Lymphocyte infiltration in blood vessels of the alveolar septum, pericardium, and liver was noted in postmortem tissue samples from COVID-19 patients [71,72,73]. The infiltration of immune cells in tissues affected by insulin fluctuations may in part contribute to the increased levels of insulin resistance in COVID-19 patients, as this may result in hyperglycemia that in turn affects disease severity [13].

A study in Finnish children, however, suggested that the increased numbers of pediatric type 1 diabetes and DKA were unlikely a direct consequence of SARS-CoV-2 infection, but rather an indirect effect of the pandemic, noted as issues related to healthcare systems and parental fears of contracting COVID-19 [74].

Although a limited number of cases have been reported so far, aforementioned findings suggest that there is a possible association between SARS-CoV-2 infection and diabetes development [75,76].

## 4. SARS-CoV-2 Receptors and Proteases in β Cells

Viruses attach to and enter host cells to start the viral life cycle by binding to receptors expressed on host cells [77]. The clinical observations described in the previous section have prompted investigation into the expression levels of SARS-CoV-2 receptors and host factors, such as proteinases that facilitate virus entry into pancreatic β cells (Table 1).

Recent single-cell RNA-sequencing (RNA-seq) analysis showed that human primary β cells express ACE2 transcripts [78,79], which was further supported by immunohistochemistry analysis of human pancreatic islets [80,81]. In contrast to these reports, analysis using publicly available RNA-seq datasets from human pancreas, pancreatic islets, and β cells showed minimal expression of ACE2 mRNA [82,83,87], which was confirmed by in situ hybridization [83]. These studies also showed that ACE2 protein was undetectable in β cells, but was found in exocrine capillaries, pericytes, the microvasculature, and in islet ducts in COVID-19 and non-COVID-19 subjects [80,82,83,87]. A potential explanation for these controversial observations is the intra- and inter-individual variations of ACE2 isoforms, and the specificity of employed antibodies to each of these isoforms [81]. There are two transcriptionally independent truncated isoforms of ACE2: an 805-aa, full-length ACE2; and a 459-aa, short-length ACE2 [88]. In vitro findings suggest that the short-length ACE2 isoform does not act as a receptor for the SARS-CoV-2 spike protein [88]. In addition, the short-length ACE2 isoform has been described as an interferon (IFN)-induced isoform, whereas the full-length protein has not [88]. However, the short-length ACE2 has been suggested to potentially interact with the full-length ACE2 isoform to modulate susceptibility to SARS-CoV-2 infection [80]. Several companies have manufactured antibodies that are either specific to the full-length isoform only, or able to bind both isoforms. Immunohistochemistry and immunoblot staining patterns in human pancreas samples can vary depending on ACE2 antibody specificity [80,83]. Other explanations for the discrepancy of ACE2 expression levels in β cells have also been discussed, such as low sample size, results based on non-COVID-19 pancreatic tissues [75,81,82,83,84], low detection sensitivity of employed methods [85,89], differences in gender and ethnic background [81,85,86], inconsistent methodology for sample preservation or preparation between studies [75], negative effects of rapid pancreas autolysis for receptor detection [82], and rapid data evaluation and fast publication turnover [75]. While the expression of ACE2 on β cells is currently controversial and further research is needed, ACE2 expression in pancreatic ducts and in the microvasculature appears to be acknowledged by some study groups [75].

Other putative SARS-CoV-2 receptors and proteases, which facilitate virus entry into β cells, have also been examined. TMPRSS2 belongs to the type II transmembrane serine protease family and is widely expressed on epithelial cells, although its physiological function remains unclear [90,91]. TMPRSS2 has two isoforms, both of which are autocatalytically activated: isoform 1, which has two N-terminal fragments; and isoform 2, which has a single N-terminal fragment [92]. TMPRSS2, which is known to proteolytically activate membrane fusion and influenza virus entry [92], has been suggested to act as an activator of the SARS-CoV spike protein once the virus has bound to ACE2 [93,94,95]. In addition, TMPRSS2 cleaves ACE2 at arginine 697–716, which enhances viral uptake [92]. However, cells lacking TMPRSS2 are still permissive to SARS-CoV-2, implying that the expression of TMPRSS2 is not essential for virus entry [85]. As with ACE2, the expression level of TMPRSS2 in β cells is controversial. Some studies showed detectable or high expression levels of TMPRSS2 protein and mRNA in pancreatic endocrine cells [81,84], whereas others did not [48,82,83]. Therefore, further study is required.

NRP-1, a member of the neuropilin family, is a type 1 transmembrane protein that plays a role in vascular endothelial growth factor and semaphorin signaling, and can also influence viral entry via host cell receptors [81] for the human T-lymphotropic virus-1 [96], Epstein–Barr virus [97], and murine cytomegalovirus [98]. NRP-1 binds to a viral surface protein that is processed by furin [99]. Recently, it was reported that NRP-1 promotes cell entry and SARS-CoV-2 infectivity [45,100,101]. SARS-CoV-2 harbors multiple arginine residues that are cleaved by furin, a process that is thought to be essential for S protein-mediated cell–cell fusion and virus entry into human lung cells [102]. Moreover, a higher NRP-1 expression was observed in pancreatic β cells compared with α cells in both non-COVID-19 subjects [48,84,85] and COVID-19 patients [84]. In addition, furin mRNA expression was detected in several pancreatic cell types, including acinar, α, β, δ, PP, ductal, and endothelial cells [85]. Ex vivo inhibition of NRP-1 in islets reduced the efficiency of SARS-CoV-2 infection, resulting in partial rescue of glucose-stimulated insulin secretion [48]. Thus, NRP-1 may contribute to the ability of SARS-CoV-2 to infect β cells [48,84,85].

CTSL, also known as CatL, is a lysosomal cysteine protease that has a role in several physiological processes, including the inflammatory response, antigen processing, apoptosis, extracellular matrix remodeling, and MHC class II-mediated immune responses [103]. Under certain conditions, CTSL is released from lysosomes into the cytosol and extracellular milieu [103,104]. A recent report found that COVID-19 patients had elevated circulating levels of CTSL, and that the levels of CTSL were associated with disease duration and severity [46]. In the context of infection, CTSL cleaves the SARS-CoV-2 S protein and enhances entry into host cells, an effect that was strongly inhibited by CTSL inhibitors [46,105]. Although CTSL expression was detected in pancreatic cells from non-COVID-19 subjects [82,83,85], the expression levels in pancreatic cells from COVID-19 patients have not been reported.

As described above, reports on the expression levels of receptors and proteases in pancreatic β cells are inconsistent; therefore, further research is needed. Nevertheless, several papers have reported that SARS-CoV-2 can infect β cells as well as other pancreatic cell types, indicating that the pancreas is one of the target organs of SARS-CoV-2.

## 5. The Distribution of SARS-CoV-2 in the Pancreas

Some studies have shown a varying distribution of SARS-CoV-2 in pancreatic autopsy samples from COVID-19 patients (Table 2).

Reports describing the type of cells and tissues that are positive for SARS-CoV-2 protein or mRNA in the pancreas of infected individuals have been inconsistent. For instance, some reports indicate that SARS-CoV-2 protein/mRNA is specifically found in insulin-positive cells [48], ductal epithelium but not endocrine tissue [83]; however, other reports indicate that both endocrine and exocrine tissue are positive for them [81,84,85,86]. The discrepancies among these reports may be partly due to differences in expression levels of virus receptors between individuals, as well as due to differences in the employed methods for pancreatic hormone and viral-antigen co-staining. Of note, reduced insulin content in islet cells upon SARS-CoV-2 infection has been reported [48,85]. Furthermore, cells staining negative for insulin but positive for β-cell lineage markers have been observed in biopsy specimens from deceased COVID-19 patients, suggesting that SARS-CoV-2 infection might hinder hormone expression in β cells [81]. Thus, insulin may not be a suitable marker for the detection of β cells in patients with COVID-19 [81]. Because insulin-specific antibodies are the most commonly used tool for pancreatic β-cell detection, other β-cell lineage-specific markers, such as NK6 Homeobox 1 (NKX6.1) [81], may be needed to accurately detect the distribution of SARS-CoV-2 in the pancreas and to determine the fate of β cells post virus infection.

Müller et al. [81] reported that SARS-CoV-2 nucleocapsid protein-positive cells were not randomly scattered across the pancreas, but instead occurred in clusters of infected cells. Therefore, the authors suggested that a localized viral spread had occurred in the pancreas, rather than a random distribution across the pancreas tissue. These observations were independent of COVID-19 disease stage (early or late) [81].

Steenblock et al. [84] detected virus-like particles in cells containing insulin secretory granules. In addition, Qadir et al. [86] demonstrated the presence of SARS-CoV-2 particles in ductal and endothelial cells of the pancreas in patients with COVID-19. These findings, which were demonstrated using electron microscopy, provide direct evidence that SARS-CoV-2 can infect pancreatic tissue.

The infiltration of cluster of differentiation (CD) 45-positive immune cells in exocrine and endocrine pancreatic tissues, which indicates pancreatic inflammation, was observed in patients with COVID-19 [84]. Conversely, others have not been able to detect any evidence of inflammation in the pancreas of individuals with COVID-19 [48,82], implying that not all COVID-19 patients display pancreatic damage despite the presence of severe COVID-19-related complications leading to death.

## 6. Pancreatic β-Cell Failure Induced by SARS-CoV-2 Infection

Literature suggests that viruses may have the potential to induce pancreatic β-cell death or damage via several mechanisms: induction of direct cell lysis, programed cell death, inflammation (bystander damage or bystander activation), autoimmunity against β cells, molecular mimicry, and dedifferentiation or transdifferentiation [17,26,27,28]. Since the COVID-19 outbreak began, researchers have been interested in the potential mechanisms leading to β-cell damage or death, which may manifest as metabolic dysfunction in patients with COVID-19.

Steenblock et al. [84] showed that only some islets from pancreas samples derived from COVID-19 patients were highly positive for phosphorylated pseudokinase mixed lineage kinase domain like (pMLKL) protein, a hallmark of necroptosis. This implies that SARS-CoV-2 infection may induce necroptotic cell death in islet cells [84], which is consistent with the report that human coronavirus triggers necroptosis in host cells [106]. Wu et al. [48] reported that human primary islets infected with SARS-CoV-2 have reduced insulin content and secretion, as well as an increased number of TUNEL-positive β cells ex vivo. Furthermore, phosphoproteomic mass spectrometry analysis suggested that activation of c-Jun N-terminal kinase (JNK)-mitogen-activated protein kinase (MAPK) apoptosis signaling is a potential mechanism leading to β-cell death post SARS-CoV-2 infection [48]. Yang et al. [78] developed human pluripotent stem cell (hPSC)-derived pancreatic endocrine-like cells to evaluate the cellular response to SARS-CoV-2. hPSC-derived pancreatic endocrine-like cells infected with SARS-CoV-2 showed upregulated transcript levels of chemokines, cytokines, and genes related to apoptotic signaling. Moreover, the infected cells had increased levels of caspase 3 protein [78], which supports the results from studies of autopsy samples and ex vivo human primary islets.

Tang et al. [85] showed that the expression of α cell and acinar cell markers were upregulated in β cells, and that insulin expression levels were downregulated, upon SARS-CoV-2 infection ex vivo. Consistent with these findings, autopsy samples from COVID-19 subjects showed a higher average intensity of the acinar cell marker, trypsin 1, in insulin-positive cells, compared with samples from control subjects. The percentage of insulin and trypsin 1 co-stained cells was also greater in samples from COVID-19 patients [85]. Müller et al. [81] found only a few number of cells positive for both insulin and SARS-CoV-2 nucleocapsid protein in pancreas samples from COVID-19 patients, whereas cells positive for both NKX6.1 (exclusively expressed in β cells) and SARS-CoV-2 nucleocapsid protein were detected close to islets or within SARS-CoV-2-infected cell clusters, suggesting that SARS-CoV-2 infection might result in reduced insulin content in β cells [81]. These data suggest that SARS-CoV-2 or COVID-19 may have the potential to induce β-cell impairment or transdifferentiation of β cells to other islet cell types.

Qadir et al. [86] reported an association between pancreatic thrombofibrosis and new-onset diabetes in patients with COVID-19. Pancreas sections from SARS-CoV-2-infected nonhuman primates (NHPs), including African green monkeys and rhesus macaques, showed multiple microthrombi in small veins throughout the pancreas, increased fibrosis, and the presence of endotheliitis with elevated levels of serum lipase compared with uninfected controls. Similar observations were found in human patients with COVID-19, including in cases of new-onset diabetes that were diagnosed upon admission. Kusmartseva et al. [83] also observed multiple thrombotic lesions in pancreas sections from patients with COVID-19. Notably, NHPs infected with SARS-CoV-2 developed diabetes 9–24 days post inoculation, which implies that long-term consequences of a fibrotic/thrombotic pancreas may indirectly lead to β-cell dysfunction and cause late-onset diabetes in patients with COVID-19 [86].

## 7. Host Factors and Viral Factors

Several factors associated with COVID-19 severity have been reported: older age, male sex, race, underlying medical conditions (comorbidities), decline of CD3-positive cells, lymphocytopenia, higher levels of interleukin (IL)-6 and IL-8, blood group A, and autoantibodies against type 1 IFNs [107,108,109,110,111,112,113,114]. In addition to these factors, genome-wide association studies and genome-wide CRISPR screens have uncovered genetic factors associated with COVID-19, as described elsewhere [110,115,116,117]. Genes, genetic loci, and single nucleotide polymorphisms affecting genes involved in type 1 IFN signaling are considered potential factors that may influence COVID-19 severity. This includes the IFN signaling molecules toll-like receptor 3 (TLR3), IFN regulatory factor 7 (IRF7), IFN-α receptor (IFNAR) 1, IFNAR2, 2′-5′-oligoadenylate synthetases (2′-5′AS), and tyrosine kinase 2 (TYK2) [118,119]. It has been suggested that impaired innate immune responses due to genetic defects may cause higher sensitivity to SARS-CoV-2 at early stages of the infection, leading to critical COVID-19 [118,119,120]. This is consistent with reports that show that patients with autoantibodies against type 1 IFNs, which abrogate innate immune responses induced by type 1 IFNs, have severe and critical COVID-19 infections [111,114]. In late stages of the disease; however, robust type 1 IFN responses due to genetic variants could potentially exacerbate the inflammatory responses that drive inflammatory organ injury, resulting in critical COVID-19 [119,120]. Therefore, this indicates that the role of type 1 IFN signaling in COVID-19 is dependent on the disease stage [121]. Genes involved in type 1 IFN signaling have also been reported as potential gene candidates that may modulate susceptibility or resistance to virus-induced diabetes, both in animal models and humans. Ingenuity pathway analysis (IPA) showed that the IFN and Janus kinase (JAK)-signal transducer and activator of transcription (STAT) signaling pathways were upregulated in ex vivo human primary islets infected with SARS-CoV-2 in comparison with mock-infected islets [85]. These results suggest that type 1 IFN signaling may play a role in SARS-CoV-2 infection of pancreatic islets.

Viral genetic factors may also contribute to and modulate the infectivity, virulence, and clinical features of SARS-CoV-2. It was reported that a D614G mutation within the spike protein of SARS-CoV-2 increases its infectivity, and this mutated form of the virus has spread faster globally than the D614 variant [122]. Of interest, a longer ORF3b variant (465 bp) of SARS-CoV-2 led to reduced ability of the virus to stimulate of IFN production in host cells compared with a shorter ORF3b variant (69 bp); moreover, this shorter variant was isolated from severe COVID-19 subjects [123]. Genetic variation in SARS-CoV-2 may alter the organ-specific infectivity and pathogenic effects in host pancreatic β cells [124]. Currently, the role of host genetic factors and viral genetic factors in the development of pancreatic β-cell failure or death in COVID-19 patients remains unclear.

## 8. Conclusions

Recent studies of COVID-19 patients indicate that SARS-CoV-2 and COVID-19 may have potential diabetogenic effects (Figure 1).

SARS-CoV-2 can potentially infect pancreatic tissue; however, the route of transmission of the virus into the pancreas remains unknown [124]. Researchers have reported several direct or indirect mechanisms of β cell failure following SARS-CoV-2 infection and COVID-19 as shown in the arrows; necroptosis [84], apoptosis [48,78], transdifferentiation [85], impairment of β cell functions [81], microthrombus [83], and thrombofibrosis [86] are some of the potential mechanisms contributing to β cell failure. The dotted line indicates the putative factors associated with β cell failure that are discussed in this article.

The pancreas, including pancreatic β cells, appear to be targets of SARS-CoV-2, and are damaged by the direct and indirect effects of infection. Further in vitro and ex vivo studies will provide more information on the mechanisms governing direct pathogen–cell interactions. However, infection elicits anti-pathogen responses in the host; therefore, to understand the complicated pathogenesis processes in the host, in vivo studies using experimental animal models and further in situ examinations of patient samples are needed. Currently, limited availability of in vivo models and samples from COVID-19 patients appear to be a major obstacle for studying the developmental mechanisms of metabolic dysfunctions caused by COVID-19. Therefore, in vivo studies using novel experimental animal models may help to determine the precise sequential changes occurring in the pancreas, as well as the developmental kinetics of diabetes, after SARS-CoV-2 infection.

Given that the interplay between host genetics and viral diabetogenicity is critical for the development of virus-induced diabetes [17,26], host genetic factors and viral genetic factors may also contribute to the development of new-onset diabetes in patients with COVID-19. Additional large-scale studies in patients and carefully designed experiments are required to determine the effect of this relationship in COVID-19 patients. Although translating the results of animal models into humans is a challenging task, novel animal models with genetic predispositions may be needed to uncover the mechanisms of β cell failure upon viral infection. These efforts will provide additional insight into the notion that viruses, as one of many environmental factors, may potentially trigger diabetes onset in people with genetic predispositions, and may reveal potential targets for interventions to prevent the development of diabetes post viral infection.

## Figures and Tables

**Figure 1 biology-11-00022-f001:**
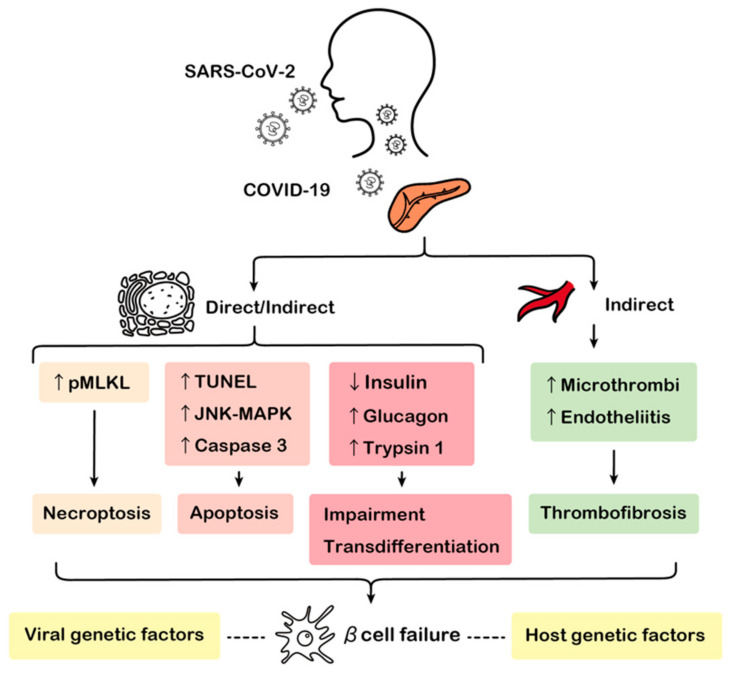
Potential mechanisms of β cell failure upon SARS-CoV-2 infection.

**Table 1 biology-11-00022-t001:** The expression of SARS-CoV-2 receptors and host factors in human pancreatic β cells.

Author [Ref.]	Samples	Methods	Receptors	Proteases, Host Factors
Yang et al. [78]	Adult human islets	Single cell RNA-seq	ACE2 (+) ^a^	TMPRSS2 (+)
	Adult human islets	Immunohistochemistry (IHC) (colocalization with insulin)	ACE2 (+)	Not tested (NT)
	Human pluripotent stem cell (hPSC)-derived insulin-positive cells	IHC (colocalization with insulin)	ACE2 (+)	NT
Liu et al. [79]	GTEx database	Bulk RNA-seq	ACE2 (+)	NT
	Pancreatic cells (NCBI GEO; GSE85241, GSE84133)	Single cell RNA-seq	ACE2 (+)	NT
Fignani et al. [80]	Human pancreas tissue section from patients with COVID-19	IHC (colocalization with insulin)	ACE2 (+)	NT
Müller et al. [81]	Adult pancreatic tissue sections from healthy subjects	IHC (colocalization with C-peptide)	ACE2 (+)	TMPRSS2 (+)
Coate et al. [82]	FACS purified human β cells (NCBI GEO; GSE67543, GSE57973)	Bulk RNA-seq	ACE2 (±) ^c^	TMPRSS2 (±)
	Adult human islets (NCBI GEO, GSE84133, GSE124742; ArrayExpress, E-MTAB-5061; HPAP Database, SCR_014393)	Single cell RNA-seq	ACE2 (±)	TMPRSS2 (±)
	Adult and juvenile pancreatic tissue sections from non-diabetic donors	IHC (colocalization with insulin)	ACE2 (−) ^b^	TMPRSS2 (−)
Kusmartseva et al. [83]	Islets from donors without diabetes or with type 2 diabetes (T2D) (NCBI GEO, GSE84133, GSE81076, GSE85241, GSE86469; ArrayExpress, E-MTAB-5061)	Single cell RNA-seq	ACE2 (±)	TMPRSS2 (±)
	Non-diabetic, SARS-CoV-2-negative human pancreata	in situ hybridization (ISH) (colocalization with insulin)	ACE2 (±)	TMPRSS2 (±)
	Non-diabetic, SARS-CoV-2-negative human pancreata	IHC (colocalization with insulin)	ACE2 (−)	NT
Steenblock et al. [84]	Human pancreas tissue section from patients with COVID-19	IHC (colocalization with insulin)	ACE2 (+), DPP4 (+)	TMPRSS2 (+), NRP1 (+)
Tang et al. [85]	Autopsy pancreas sample of non-COVID-19 subjects	IHC (colocalization with insulin)	NT	NRP1 (+)
	Primary human islets infected with SARS-CoV-2 ex vivo	Single cell RNA-seq	NT	FURIN (+), CTSL (+)
Qadir et al. [86]	Pancreatic sections from male and female human donors	IHC (colocalization with insulin)	ACE2 (±)	NT
Wu et al. [48]	Adult pancreatic tissue sections from healthy subjects	IHC (colocalization with insulin)	ACE2 (±)	TMPRSS2 (±), NRP1 (+), TFCR (+)

^a^ (+), positive; ^b^ (−), negative; ^c^ (±), positive but low level.

**Table 2 biology-11-00022-t002:** The distribution of SARS-CoV-2 in pancreatic tissue from patients with COVID-19.

Author [Ref.]	Methods	Targets	Targets-Positive Cells	Samples	Donor Clinical Information Associated with Diabetes	Number of Analyzed Samples	Donor Race/Ethnicity	Other Findings from Pancreas Sections and Islets Infected with SARS-CoV-2 Ex Vivo
Wu et al. [48]	IHC (colocalization with insulin)	SARS-CoV-2 nucleocapsid protein (NP) SARS-CoV-2 spike protein (SP)	Selective to insulin (+) cells in autopsy samples Primarily in insulin (+) cells in ex vivo experiments	Pancreatic autopsy specimens from COVID-19 patients Non-COVID-19 human pancreatic islets infected with SARS-CoV-2 ex vivo	One COVID-19 patient had a history of T2D (1/9). None of the islet donors (non-COVID-19 subjects) had a history of diabetes.	COVID-19, *n* = 9; non-COVID-19, *n* = 13	Not available (NA)	Islets infected with SARS-CoV-2 ex vivo have reduced insulin content and secretion, as well as an increased number of TUNEL-positive β cells.
	ISH (colocalization with insulin)	SARS-CoV-2 spike mRNA	Insulin (+) cells	Pancreatic autopsy specimens from COVID-19 patients	One COVID-19 patient had a history of T2D (1/4).	COVID-19, *n* = 4		
Kusmartseva et al. [83]	IHC (hematoxylin counterstain)	SARS-CoV-2 NP	Ductal epithelium cells	Pancreatic autopsy specimens from COVID-19 patients	Two COVID-19 patients had a history of T2D (2/3).	COVID-19, *n* = 3	Caucasian, African American	Multiple thrombotic lesions were observed in pancreatic ducts.
Steenblock et al. [84]	IHC (colocalization with insulin)	SARS-CoV-2 NP	Endocrine and exocrine	Pancreatic autopsy specimens from COVID-19 patients and non-COVID-19 subjects	None of the COVID-19 patients, except for one patient, had a history of T2D (1/11). None of the non-COVID-19 subjects, except for one subject, had a history of T2D, or no data were available (1/5).	COVID-19, *n* = 11; non-COVID-19, *n* = 5	NA	The hallmark feature of necroptosis was observed in pancreatic endocrine and exocrine cells in patients with COVID-19.
	Electron microscopy	Virus particles	Cells containing insulin secretory granules	Pancreatic autopsy specimens from COVID-19 patients	None of the COVID-19 patients had a history of diabetes.	COVID-19, *n* = 1		
Müller et al. [81]	IHC (colocalization with insulin, NKX6.1, cytokeratin 19 (CK19))	SARS-CoV-2 NP	Small ducts (CK19 (+) cells), acinar cells, insulin (+) cells (a small number), close to islets, NKX6.1 (+) cells	Non-COVID-19 human pancreatic islets infected with SARS-CoV-2 ex vivo Pancreatic autopsy specimens from COVID-19 patients	All COVID-19 patients had comorbidities including hypertension (HT), chronic kidney disease (CKD), coronary heart disease, obstructive pulmonary disease, aortic valve stenosis, and cancer. None of the non-COVID-19 subjects had a history of diabetes.	COVID-19, *n* = 4; non-COVID-19, *n* = 4	NA	Only a few cells positive for both insulin and SARS-CoV-2 NP were found in pancreas samples from COVID-19 patients, whereas cells positive for both NKX6.1 (exclusively expressed in β cells) and SARS-CoV-2 NP were detected in higher amounts.
	Electron microscopy	Virus particles	Dilated Golgi vacuoles	Non-COVID-19 human pancreatic islets infected with SARS-CoV-2 ex vivo	None of the islet donors (non-COVID-19 subjects) had a history of diabetes.	Non-COVID-19, *n* = 2		
Qadir et al. [86]	IHC (colocalization with insulin, CD31, CK19)	SARS-CoV-2 NP	Insulin (+) cells, non-insulin (+) islet cells, CD31 (+) endothelial cells, CK19 (+) ductal cells	Pancreatic autopsy specimens from COVID-19 patients	Three patients did not have a history of diabetes (3/5), two of whom presented with non-fasting glucose (NFG) levels >300 mg/dL at admission. Two other patients had T2D (2/5).	COVID-19, *n* = 5	Hispanic, non-Hispanic white, non-Hispanic black	Multiple microthrombi in the pancreas venous and increased fibrotic area in pancreas sections were observed in patients with COVID-19.
	Electron microscopy	Virus particles	Ductal cells and endothelial cells	Pancreatic autopsy specimens from COVID-19 patients	No diabetes history (NFG >300 mg/dL at admission).	COVID-19, *n* = 1	Non-Hispanic white	
Tang et al. [85]	IHC (colocalization with insulin, E-cadherin, CD31, CK19, trypsin 1, vimentin)	SARS-CoV-2 NP	Insulin (+) and E-cadherin (+) cells, CD31 (+) endothelial cells, CK19 (+) ductal cells, trypsin 1 (+) acinar cells, vimentin (+) mesenchymal cells	Pancreatic autopsy specimens from COVID-19 patients and non-COVID-19 subjects	One patient had no comorbidities (1/5); the other patients had chronic comorbidities including T2D, dementia, HT, and atrial fibrillation (4/5).	COVID-19, *n* = 5; non-COVID-19, *n* = 8	NA	The islets infected with SARS-CoV-2 ex vivo showed reduced expression levels of insulin and increased expression levels of glucagon.
	Single cell RNA-seq	SARS-CoV-2 viral RNAs (SARS-CoV-2-E, SARS-CoV-2-M, SARS-CoV-2-ORF1ab, SARS-CoV-2-ORF8, SARS-CoV-2-ORF10, and SARS-CoV-2-S)	Highly expressed in PRSS1 (+) acinar cells, GCG (+) α cells, INS (+) β cells, KRT19 (+) ductal cells, and COL1A1 (+) fibroblasts Expressed at relatively low levels in PYY (+) PP cells, SST (+) δ cells, PECAM1 (+) endothelial cells, and LAPTM5 (+) immune cells	Non-COVID-19 human pancreatic islets infected with SARS-CoV-2 ex vivo	The islet donors (non-COVID-19 subjects) did not have a history of diabetes.	Non-COVID-19, *n* = 2		

## Data Availability

Not applicable.

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
