# Peer review of "SARS-CoV-2 Infection and Pancreatic β Cell Failure"

_biology, 2021, doi:10.3390/biology11010022_

Round 1
Reviewer 1 Report
In this article, Mine and colleagues discuss a possible link between SARS-CoV-2 infection and pancreatic b-cell failure, leading to diabetes. In my opinion, the article should be adjusted to better reflect the literature and broad its target audience. Maybe expand the scope of the review to also talk about SARS-CoV-2 effects in organ-tissues affected by insulin fluctuations, such as liver, skeletal muscle, vasculature, and heart. It is also important to highlight the idea that COVID-19 may worsens or leads to new onset diabetes, as well as potential implications post-SARS-CoV-2 infection. Authors should also mention and create a graphical representation of likely mechanisms involved in this process would also improve the overall impact of the manuscript. In summary, while the idea is of interest, the manuscript as it stands is only a descriptive overview of the literature (evidenced by both tables), and not a topical summary of the material published by the scientific community. The paper should also be proofread.
Author Response
We appreciate the time and effort that you have dedicated to providing your valuable feedback on our manuscript.
Please see the attachment.

Reviewer 2 Report
This manuscript reviews the potential relationships between COVID-19 and pancreatic beta cell failure linked to the development of diabetes. Initially, the authors give a short description of SARS-CoV-2 characteristics and provide a literature review of the various, often controversial, reports on the association between COVID-19 and diabetes. Suggesting that the association is possible (mainly supported by the evidence of Section 6), the authors summarize the existing results on the distribution of SARS-CoV-2 in pancreas and discuss in depth the potential associations between COVID-19 and beta cell failure.
I would like to thank the authors for this comprehensive review and the information extracted from the limited available studies on this subject. I would like to ask them to please address the below comments:
- The studies highlighted at Table 2 could have been discussed in more detail. Could the authors please indicate the experimental design of each study? Also, what are the pros and cons of each study relative to the claims that they make? What are the other main results apart from the SARS-CoV-2 positive cells (if any)?
- Section 7 summarizes some reported factors associated with COVID-19 severity: age, gender, medical conditions and other immune related characteristics. Is race among these factors? Is the demographic / clinical information available in the studies highlighted at Table 2? If yes, I would like to ask the authors to also summarize this information, as well as the number of subjects analyzed, per study.
- In the Conclusion, the authors stress the need for conducting systematic studies to extract more reliable information on the association between COVID-19 and diabetes. I agree with this statement, but I would like to see more details on what is missing from the current studies (e.g. small samples sizes etc), what should be improved and why (technology, experimental design etc) and the authors’ potential plans or suggestions (e.g. why using a particular technology, what we gain from it, what experimental design is needed etc). In the current form, the conclusion only slightly addresses these important issues.
- Have the authors generated any preliminary data related to this study and, if yes, do the findings support the current evidence?
Author Response

(The authors gave the same response as above.)

Round 2
Reviewer 1 Report
No further comments.
Author Response
Dear Reviewer
We wish to express our appreciation to the Reviewer for your valuable feedback, which has greatly helped us to improve the quality of our manuscript.
Sincerely,
Keiichiro Mine
Reviewer 2 Report
Dear authors,
Thank you for addressing my comments. The manuscript has been greatly improved. The updated Table 2 is very informative (clinical information, experimental design etc) and highlights some pros and cons of the existing studies. I agree with your conclusions about in vivo samples and genetic pre-disposition information. Based on the data of Table 2 and your conclusions I would also add "carefully designed experiments" and, regarding animal models, "translation of the results of animal models to humans" which for Diabetes (among others) is a challenging task.
Some minor comments for Tables 1 and 2:
- You could probably remove the Ref column (last) and include the reference number as superscript at the authors' name of the first column (to save space).
- Please use smaller fonts in Table 2 for the sentences to occupy less lines that make the table more readable (e.g. shorten the sentences, replace 'type 2 diabetes' with T2D, 'hypertension' with HT, 'immunohistochemistry' with IHC etc)
- Other minor edits required: few floating dots, commas and words
